

# Comparison of microbial community structures in soils with woody organic amendments and soils with traditional local organic amendments in Ningxia of Northern China

Zhigang Li[1,*], Kaiyang Qiu[1,*], Rebecca L. Schneider[2], Stephen J. Morreale[2] and Yingzhong Xie[1]

[1] School of Agriculture, Ningxia University, Yinchuan, Ningxia, China
[2] Department of Nature Resources, College of Agriculture and Life Sciences, Cornell University, Ithaca, NY, USA
* These authors contributed equally to this work.

Corresponding author
Yingzhong Xie, xieyz@nxu.edu.cn, yzh.xie@iCloud.com

## ABSTRACT

**Background**. Addition of organic amendments has been commonly adopted as a means to restore degraded soils globally. More recently, the use of woody organic amendments has been recognized as a viable method of capturing and retaining water and restoring degraded and desertified soil, especially in semi-arid regions. However, the impacts of woody amendments on soil microbial community structure, versus other traditional organic supplements is less understood.

**Methods**. Three locally available natural organic materials of different qualities, i.e., cow manure (CM), corn straw (CS), and chipped poplar branches (PB) were selected as treatments in Ningxia, Northern China and compared with control soils. Four microcosms served as replicates for each treatment. All treatments contained desertified soil; treatments with amendments were mixed with 3% (w/w) of one of the above organic materials. After 7 and 15 months from the start of the experiment, soil samples were analyzed for chemical and physical properties, along with biological properties, which included microbial $\alpha$-diversity, community structure, and relative abundance of microbial phyla.

**Results**. Both bacterial and fungal $\alpha$-diversity indices were weakly affected by amendments throughout the experimental period. All amendments yielded different microbial community compositions than the Control soils. The microbial community composition in the CS and PB treatments also were different from the CM treatment. After 15 months of the experiment, CS and PB exhibited similar microbial community composition, which was consistent with their similar soil physical and chemical properties. Moreover, CS and PB also appeared to exert similar effects on the abundance of some microbial taxa, and both of these treatments yield different abundances of microbial taxa than the CM treatment.

**Conclusion**. New local organic amendment with PB tended to affect the microbial community in a similar way to the traditional local organic amendment with CS, but different from the most traditional local organic amendment with CM in Ningxia, Northern China. Moreover, the high C/N-sensitive, and lignin and cellulose

decompose-related microbial phyla increased in CS and PB have benefits in decomposing those incorporated organic materials and improving soil properties. Therefore, we recommend that PB should also be considered as a viable soil organic amendment for future not in Ningxia, but also in other places.

## INTRODUCTION

The incorporation of locally available organic materials into degraded soil has been a widely adopted practice. Organic amendments are recognized for their roles in increasing soil water holding capacity, soil porosity, and water infiltration and percolation, while decreasing soil crusting and bulk density (*Thangarajan et al., 2013*). Moreover, organic amendments stimulate soil microbial community growth and activity in degraded soils, resulting in mineralization of nutrients available to plants, and increasing soil fertility and quality (*Luna et al., 2016*). Based on their benefits of improving water status and soil fertility, organic amendments are extensively used in dry and desertified soils globally. In the arid and semi-arid regions of China and elsewhere, retaining and returning crop residues (e.g., wheat straw and corn straw) and adding livestock manure (e.g., cow manure, chicken manure, pig manure) to the soil have long been regarded as a practical and effective method for improving soil quality and crop productivity (*Fan et al., 2014*). However, a seldom employed strategy in these regions is the use of woody materials as a soil amendment technique, although there is much available materials that could be readily collected from more than 2.2 million km² of shelterbelt forests extending across China (*Dai & Chu, 2010*). More specifically, rapidly growing poplars (Populus *alba* and other members of the genus) are the most widespread and available tree throughout China (*Jia et al., 2013*). Moreover, in the highly degraded and desertified areas of Northern China, three-quarters of the poplar trees are planted as farmland shelterbelts (*Jing, Xing & Du, 2015*). Within the seriously desertified Ningxia Autonomous Region, poplar species rank first among all trees and distribute through an area of 47,000 km² (*Sun et al., 2009*).

Notably, our previous research has demonstrated that incorporation of as little as 2% wood chips (w/w) into the surface soil can effectively capture and retain more precipitation and improve soil physicochemical properties in Ningxia, China (*Li et al., 2018*). Additionally, we reported that wood chips presented several comparable benefits to traditional local amendments, such as crop straw and cow manure, in improving soil health and enhancing crop growth. Importantly, in contrast to rapidly decomposing compost materials, the incorporation of woody material in severely degraded soils can have valuable and long-lasting benefits due to its low decomposition rate (*Weedon et al., 2009*). However, the effects of wood chips as well as other traditional local organic amendments on soil microbial communities are insufficiently understood. Microbial communities are important for the functioning of the ecosystem, both in relation to
direct interactions with plants, and with regard to nutrient and organic matter cycling (*Adak & Sachan, 2009*). Moreover, microbial abundance, activity, and composition largely determine the sustainable productivity of agricultural land (*Heijden, Bardgett & Straalen, 2008*), since microbes play an integral and essential role in all soil processes (*Barrios, 2007*). In particular, soil microbial communities play an important role in the C cycle and the metabolism of other nutrients, such as nitrogen for energy acquisition (*Falkowski, Fenchel & Delong, 2018*). Some scientists have reported that C addition seems to select for specific microbial groups that feed primarily on organic compounds, changing the composition of the microbial community (*Zhong et al., 2010*; *Hu et al., 2010*). But other studies also indicated that a change in microbial community structure does not always involve a change in microbial community function or an increase in availability of plant nutrients and crop productivity (*Franco-Otero et al., 2012*; *Lazcano et al., 2013*). Nevertheless, a series of studies have focused on exploring the functions of microorganism in soil process, and they have reported the ecological roles some microorganisms are playing in soils, including C, N and other nutrient cycling (*Fierer, Bradford & Jackson, 2007*; *Fierer et al., 2012*; *Hartmann et al., 2015*; *Zhang et al., 2018*). These previous studies are useful for us to speculate what ecological roles microorganisms may play in our soil organic amendments, because wood decomposition processes are related to microorganisms in the soil (*Sariyildiz, Anderson & Kucuk, 2005*) and those microorganisms, in turn, affect soil nutrient cycling (*Wutzler et al., 2017*). In addition, soil microbial parameters can react rapidly to changes in soil management (*Gil-Sotres et al., 2005*), and shortly after application of organic amendments, fast increases in soil microbial biomass and activity have been observed (*Dinesh et al., 2010*). For all of these reasons, it is important for us to assess short-term effects of our soil organic amendments via the reaction of microbial communities.

Previous studies also have indicated that pH and organic C were two major soil properties that drive soil microbial communities (*Hollister et al., 2010*; *Liu et al., 2014*). In this context, we assume that exogenous C input, such as organic amendments, would substantially influence microbial communities in soils. Moreover, woody materials and crop straw have more lignin and carbon (C) content, and a higher C: nitrogen (N) ratio than livestock manures, which makes them more difficult to decompose, so we hypothesize that they also differently affect soil microbes relative to manure, and other short-lived organic amendments. This contention is supported in other studies that indicated that organic amendments with a high C:N ratio resulted in higher microbial biomass and activity, and also displayed differences in microbial composition, compared to materials with low C:N ratio (*Heijboer et al., 2016*). Nevertheless, we know little about the effects of all local organic amendments, such as cow manure, corn straw, and poplar branches, on soil microbiota in degraded and desertified soils such as those in Ningxia, China. Better understanding of the microbial processes that take place in soils that have been treated with organic amendments could help identify the main drivers determining nutrient availability in order to improve soil health and facilitate crop growth. Therefore, in this context, in pursuit of the overall goal of our study, we employed an Illumina sequencing approach of bacterial and fungal ribosomal markers to compare the effects on soil microbiota of the addition of new woody amendments and the addition of traditional local amendments. The detailed objectives of

this study were to (a) compare soil amendments of poplar chips versus traditional organic amendments, including cow manure and corn straw, and their impacts on soil microbial $\alpha$-diversity and community composition, (b) reveal the different effects of local organic amendments on specific microbial taxa; (c) determine the relationships between soil properties and soil microbial $\alpha$-diversity, community composition and specific microbial taxa, respectively.

## MATERIAL & METHODS

### Study site

A microcosm experiment was conducted for 15 months from April 2015 to July 2016 in a greenhouse in the Yinchuan Belly Desert, which is located at the eastern foot of the Helan Mountains in Ningxia Hui Autonomous Region, China (106°08′~107°22′E, 38°28′~38°42′N). The characteristics of the study site are representative of those associated with the ongoing desertification processes throughout Northern China. Historically, the native landscape was grasslands, but it has been desertified after centuries of agriculture. Currently, the region is characterized by shifting dunes with scattered farmland shelterbelt forests, which along with the farmlands are supported by extensive irrigation. The elevation is 1,115 m above mean sea level and the region is characterized by a temperate continental climate, with 181.2 mm average annual precipitation and 1,882.5 mm mean annual evaporation. The annual average temperature is 10.1 °C with a maximum of 37.2° in July, and a minimum of −27.9 °C in January. The average wind speed is 1.6 m s$^{-1}$ and the frostless period is 160–170 days each year. The particle size distribution of soils is 92.5% sand (size < 2 mm), 5.5% silt, and 2.0% clay. The soil chemical properties are listed in Table S1.

### Material collection and preparation

The base soils for this experiment were collected at the study site. After collection, soils were sifted through a sieve with a mesh diameter of 2 mm and were air-dried before mixing with the selected organic materials or being used as control soils. Woody materials were obtained from pruned branches from a locally abundant poplar species (*Populus alba* L.), and herbaceous materials consisted of straw residues collected from local corn (*Zea mays* L.) after harvesting. Poplar branches and corn straw were air-dried, then ground to lengths of about 0.5 cm. Cow manure, collected from a local dairy, was also air-dried and sieved through a screen with a mesh diameter of 0.5 cm to remove coarse material. The chemical properties of organic materials are also listed in Table S1.

### Experimental design

Four treatments were tested in this study, including (1) the Control with desertified soil only, and soil treatments with incorporated organic material (3% w/w), including: (2) 3% cow manure (CM); (3) 3% corn straw (CS); and (4) 3% poplar branch (PB). Organic materials were added to the base soil and then mixed fully. For example, the soil mixture of 3% corn straw was created by adding 300 g corn straw (dry weight) to 9,700 g base soil (dry weight) and mixing completely. Within the continuum of amendments, these proportions

represented the minimum requirements for soil water retention, which was tested in a previous set of our experiments (*Li et al., 2018*).

Control soil and soil mixtures were placed in buckets with top and bottom diameter of 31.5 cm and 26.0 cm, respectively, a height of 31.5 cm. All buckets had small drainage holes at the bottom. Twelve kilograms of each soil mixture (base soil + 3% incorporated material) was packed in each bucket, after which 5 g of urea ($CO(NH_2)_2$) and 2 g of monopotassium phosphate ($KH_2PO_4$), dissolved into solution, were added to each bucket prior to the experiment. In order to supply basic nutrient requirement for alfalfa growth, the required amounts of urea and monopotassium phosphate added to the bucket were based on the results of our previous studies (*Li et al., 2018*). Initially, 12 plump alfalfa seeds were sowed in each bucket, but after the first month, only the six most robust seedlings were kept growing for the remainder of the experiment. Alfalfa was selected because it is an indicator plant in our study region. In all, 4 treatments × 4 replicates (buckets) × 2 times (7 months and 15 months) = 32 buckets were set up and arranged in a randomized block design in a greenhouse. Throughout the whole experiment, soil moisture of all buckets was monitored based on weight. All buckets were watered once every ten days and were maintained at an approximately constant water content equal to about 70% of the water holding capacity of the Control. The indoor greenhouse temperature was maintained at 15–20 °C from November to the following March, and 20–35 °C, from April to October during the experimental period.

## Soil sampling and nutrient measurements

All 32 buckets were divided into two successive batches with 16 buckets, including four treatments with four replicates, in each batch. For each batch, the length of time before the measurements was based on the precise alfalfa flowering stage. The analyses of the first batch of 16 buckets were conducted when the alfalfa first flowered and was harvested in October 2015 (seven months after the start of the experiment), while the second set of measurements were conducted in July 2016 (15 months after the beginning of the experiment) when the alfalfa was harvested after flowering for a second time. After plants were harvested, soil from each entire bucket was mixed with a sterile shovel. Then a portion was collected, air dried, and sieved through a sieve with the mesh diameter of 0.5 mm for chemical analysis, while another portion was placed in a sterile plastic bag, transported to the laboratory on ice, and then sieved (2-mm-mesh sieve) and stored at −80 °C for DNA extraction and later analysis of other biological property.

Soil organic C (SOC) was measured with the $K_2Cr_2O_7$–$H_2SO_4$ oxidation method of Walkley-Black (*Nelson et al., 1996*); total N (TN) was analyzed using the Kjeldahl procedure (*ISSCAS, 1978*); Alkaline-hydrolyzable (AN) was determined by an alkaline diffusion method (*ISSCAS, 1978*); total P (TP) was determined using $H_2SO_4+HClO_4$ digestion (*Olsen, Sommers & Page, 1982*); available P (AP) was extracted with 0.5 mol $l^{-1}$ $NaHCO_3$ (pH 8.5) (*ISSCAS, 1978*); total K (TK) was determined following the Cornfield method (*Kundsen, Peterson & Pratt, 1982*); and available K (AK) was determined using a $CH_3COONH_4$ extraction method (*Tran & Simard, 1993*). Microbial C (MBC) and N

(MBN) were estimated by the fumigation-extraction method (*Vance, Brookes & Jenkinson, 1987*). Above data were listed in Table S2.

## DNA extraction, PCR amplification, and Illumina sequencing

Samples were separately tested at each sampling date. The total DNA was extracted from 0.5 g homogenized soil per sample using a Power Soil™ DNA Isolation Kit (MOBIO Laboratories, Carlsbad, CA, USA) following the manufacturer's instructions. A total of 16 extracted DNA samples, at each sampling date, were quantified using a Nanodrop-1000 spectrometer (NanoDrop Technologies, Wilmington, DE, USA) and diluted to a final concentration of 10 ng $\mu L^{-1}$. We used the universal primers 515F5′-GTGCCAGCMGCCGCGGTAA -3′ and 909R 5′-CCCCGYCAATTCMTTTRAGT -3′ that target the V4-V5 regions of the 16S rRNA bacterial gene and produce accurate phylogenetic information (*Weisburg et al., 1991*). For fungi, the ribosomal ITS region was amplified using primers ITS1F: 5′-CTTGGTCATTAGAGGAAGTAA-3′ (*Gardes & Bruns, 1993*) and ITS2R: 5′-GCTGCGTTCTTCATCGATGC-3′ (*White et al., 1990*). To simultaneously analyze several samples in a single sequencing run, the 5′ end of the forward primer was fused with a 12 bp different barcode sequence to a single sample. Each 25 µL PCR reaction contained 10 ng extracted DNA as a template, $Mg^{2+}$-free PCR buffer, 3 mM $MgCl_2$, 200 mM dNTP, 200 nM forward primer, 200 nM reverse primer, and 1 unit of PrimeSTAR Max DNA Polymerase (Takara, Dalian, China). The thermal cycling protocol was 95 °C for 2 min as the first step, followed by 30 cycles of PCR at 95 °C for 15 s, at 57 °C for 15 s, and at 72 °C for 40 s, and a final 10-min extension at 72 °C. All the amplicons were run in a 1% agarose gel (w/v), and the separated bands of the appropriate length were excised from the gel and purified using a QIAquick Gel Extraction kit (QIAGEN GmbH, Hilden, Germany). Finally, high-throughput sequencing of the 16S rDNA and ITS rDNA genes were conducted using Illumina MiSeq, and 300 bp paired-end reads were generated (Nanjing Tree & Cloud InfoTech Ltd., Nanjing, China). All the sequences have been deposited in the Sequence Read Archive (SRA) of NCBI under the accession number of PRJNA540407 for 16S rDNA and the accession number of PRJNA540415 for ITS rDNA.

## Post-run analysis for 16S rDNA

The raw data were processed using the UPARSE pipeline with default parameters for each of the following steps: (1) to sort those exactly matching the specific barcodes into different samples; (2) to merge the paired-end reads with FLASH; (3) to trim the adapters, barcodes, and primers, and to remove sequences shorter than 200 bp; (4) to perform a quality filter using E (the sum of the error probabilities) >1; and (5) to cluster operational taxonomic units (OTUs) using a 97% identity threshold, discard singleton reads, and perform chimera filtering (http://drive5.com/uparse/) (*Edgar, 2013*). However, in order to compare relative difference between samples in downstream analyses, a randomly selected subset of 10,510 sequences were considered per bacteria community sample and 7,786 sequences per fungal community sample.

## Statistical analysis

Indices of $\alpha$-diversity analysis (Shannon and Simpson indices) were carried out in Quantitative Insights Into Microbial Ecology (QIIME) on rarefied OTU tables (Dataset S1), which were obtained from the results of the high-throughput sequencing. Then, all data analyses were computed using the vegan package in R (*Oksanen et al., 2018*). In all tests, a *p*-value <0.05 was considered statistically significant. The differences of soil properties, microbial $\alpha$-diversity, and relative abundances of microbial taxa among amendments were tested using one-way ANOVA, followed by Tukey's post-hoc test. The statistical significances of the dissimilarities in the microbial community composition between all pairs of amendments were analyzed with permutational multivariate analysis of variance. Principal coordinate analysis (PCoA) based on the Bray–Curtis dissimilarity was used to visualize the distribution patterns of the microbial community. Distance-based redundancy analysis (dbRDA) (based on the Bray–Curtis dissimilarity) was used to estimate the proportion of variability in the microbial community composition caused by each of the selected soil properties, and marginal tests were performed to test the significance of each test. Pearson's correlation analysis was used to detect the relationships between soil properties and the means of $\alpha$-diversity, as well as soil properties and relative abundance of microbial taxa.

# RESULTS AND DISCUSSION

## Effects of local organic amendments on microbial $\alpha$-diversity

It seems that previous studies have not achieved an agreement on the effect of organic amendments on soil microbial $\alpha$-diversity. Some studies reported an increase (*Francioli et al., 2016*; *Sharma et al., 2017*), some studies found a decrease (*Montiel-Rozas et al., 2018*), and some others found no effect on soil microbial $\alpha$-diversity after organic amendment (*Zhang et al., 2018*). Our study appears similar to the latter, in that we found organic amendments had weak effects on microbial $\alpha$-diversity, although there were some significant differences observed for in Shannon index values for fungi after 7 months (Table 1). Specifically, only the CM treatment had significantly lower fungal Shannon index values than CS after 7 months; there were no significant differences between other treatments. All other bacterial and fungal $\alpha$-diversity indices were not significantly influenced by amendment both after 7 months and after 15 months. Other researchers have concluded that it's difficult to draw robust inferences from the effect of organic amendments on soil microbial $\alpha$-diversity, in part because these metrics often have little power in explaining differences in community structure (*Hartmann & Widmer, 2006*). Other studies also confirmed that microbial diversity is a highly complex parameter, and its measurement by diversity indices is usually less informative than qualitative community structure analysis (*Bonilla et al., 2012*). Our studies support those studies.

Similarly, the correlation analysis, between soil properties and microbial $\alpha$-diversity, also indicated that organic amendments had weak effects on $\alpha$-diversity, since there were only few soil properties that were significantly correlated based on Shannon or Simpson index values (Table 2). Nevertheless, we should acknowledge that some other soil properties that

**Table 1** Effects of organic amendments on $\alpha$-diversity of soil bacteria and fungi after durations of 7 and 15 months (mean ± SE).

| Time | Treatment | Bacteria | | Fungi | |
|---|---|---|---|---|---|
| | | Shannon | Simpson | Shannon | Simpson |
| 7 months | Control | 8.07 ± 0.62a | 0.94 ± 0.03a | 3.94 ± 0.48ab | 0.77 ± 0.08a |
| | CM | 7.96 ± 0.24a | 0.98 ± 0.00a | 3.58 ± 0.06b | 0.76 ± 0.02a |
| | CS | 8.96 ± 0.12a | 0.99 ± 0.00a | 4.83 ± 0.23a | 0.89 ± 0.02a |
| | PB | 8.22 ± 0.22a | 0.99 ± 0.00a | 4.25 ± 0.05ab | 0.89 ± 0.01a |
| 15 months | Control | 8.97 ± 0.14a | 0.99 ± 0.00a | 4.38 ± 0.34a | 0.90 ± 0.02a |
| | CM | 8.75 ± 0.35a | 0.99 ± 0.01a | 5.46 ± 0.26a | 0.95 ± 0.01a |
| | CS | 9.17 ± 0.11a | 0.99 ± 0.00a | 3.32 ± 0.82a | 0.68 ± 0.14a |
| | PB | 9.11 ± 0.10a | 0.99 ± 0.00a | 4.77 ± 0.61a | 0.91 ± 0.03a |

**Notes.**

CS, corn straw; CM, cow manure; PB, poplar branch.

Means with different letters are significantly different with $p < 0.05$ assessed by Tukey's HSD test.

we did not determine in this study might impact $\alpha$-diversity as well. Meanwhile, we should also acknowledge that alfalfa growth affected microbial $\alpha$-diversity.

## Effects of local organic amendments on microbial community composition

Previous studies have indicated that even when microbial $\alpha$-diversity was unaffected, organic amendments have shown a strong influence on the soil microbial community composition (*Bonilla et al., 2012*). Similarly in this study, all local organic amendments exhibited different microbial community compositions than the Control at after both 7 and 15 months of study ($p < 0.05$) (Table 3). In addition, both CS and PB treatments had different bacterial and fungal community compositions than CM at both sampling times, which was probably linked to the different soil properties among these treatments (Table S2). However, the CS treatment exhibited different bacterial and fungal community composition than the PB treatment after seven months, but they had similar microbial community compositions after 15 months ($p > 0.05$). Reinforcing the link with soil properties, treatments CS and PB had dissimilar soil properties after seven months, but they became more similar after 15 months. For example, AK was higher in the CS treatment than in PB after seven months, but showed no significant difference by the time of the second sample. Similar findings were also documented by others (*Zhang et al., 2018*).

In our study, PCoA analysis also showed that microbial communities in all treatments were only weakly separated from each other after seven months (Fig. 1A and Fig. 1C). However, after 15 months, all communities in amendment treatments were clearly separated from the Controls, and CS and PB were clustered together for both the bacterial and fungal communities and separate from the CM and Control treatments (Fig. 1B and Fig. 1D).

In all, compared with the Control, all local organic amendments altered the microbial community composition, but in the non-traditional PB organic amendment, microbial community composition became similar to the traditional CS amendment only at a later stage. Notably, both CS and PB amendments had different effects on microbial

Li et al. (2019), *PeerJ*, DOI 10.7717/peerj.6854

**Table 2  Correlations between microbial α-diversity and soil properties.**

| Time | Microbiome | α-diversity | pH | SOC | TN | TP | TK | AN | AP | AK | MBC | MBN |
|---|---|---|---|---|---|---|---|---|---|---|---|---|
| 7 months | Bacteria | Shannon | 0.273 | 0.160 | −0.121 | −0.083 | −0.111 | −0.421 | −0.247 | −0.014 | 0.353 | 0.358 |
| | | Simpson | 0.232 | 0.164 | −0.261 | −0.446 | −0.314 | −0.564[*] | −0.449 | −0.232 | 0.634[**] | 0.430 |
| | Fungi | Shannon | 0.164 | 0.340 | −0.064 | −0.287 | −0.184 | −0.348 | −0.325 | −0.185 | 0.649[**] | 0.480 |
| | | Simpson | 0.302 | −0.198 | −0.160 | −0.350 | 0.055 | −0.293 | −0.378 | −0.229 | 0.256 | −0.111 |
| 15 months | Bacteria | Shannon | −0.026 | −0.108 | 0.156 | 0.467 | −0.248 | 0.418 | 0.464 | 0.609[*] | −0.284 | −0.188 |
| | | Simpson | 0.273 | 0.160 | −0.121 | −0.083 | −0.111 | −0.421 | −0.247 | −0.014 | 0.353 | 0.358 |
| | Fungi | Shannon | 0.231 | 0.627[**] | 0.367 | 0.184 | 0.319 | −0.266 | 0.114 | 0.294 | 0.362 | 0.522[*] |
| | | Simpson | 0.164 | 0.340 | −0.064 | −0.287 | −0.184 | −0.348 | −0.325 | −0.185 | 0.649[**] | 0.480 |

**Notes.**

[*] $p < 0.05$.

[**] $p < 0.01$.
**Table 3 Pairwise comparison of soil microbial community composition under different organic amendments at different times.** Pairwise comparisons were analyzed by multivariate permutational analysis of variance (PERMANOVA). Values represent the pseudo- $F$ ratio ($F$) and the level of significance ($p$).

| Pairwise comparison | 7 months | | | | 15 months | | | |
|---|---|---|---|---|---|---|---|---|
| | Bacterial | | Fungi | | Bacterial | | Fungi | |
| | $F$ | $p$ | $F$ | $p$ | $F$ | $p$ | $F$ | $p$ |
| Control vs CM | 3.21 | 0.030 | 2.99 | 0.035 | 3.81 | 0.032 | 4.55 | 0.034 |
| Control vs PB | 3.24 | 0.031 | 3.64 | 0.032 | 2.04 | 0.033 | 3.79 | 0.024 |
| Control vs CS | 2.58 | 0.030 | 4.14 | 0.029 | 2.60 | 0.033 | 4.01 | 0.023 |
| CM vs PB | 7.97 | 0.024 | 3.57 | 0.023 | 3.17 | 0.031 | 6.77 | 0.025 |
| CM vs CS | 7.83 | 0.038 | 2.91 | 0.028 | 3.57 | 0.034 | 3.96 | 0.032 |
| PB vs CS | 2.87 | 0.037 | 3.36 | 0.036 | 1.42 | 0.185 | 1.69 | 0.062 |

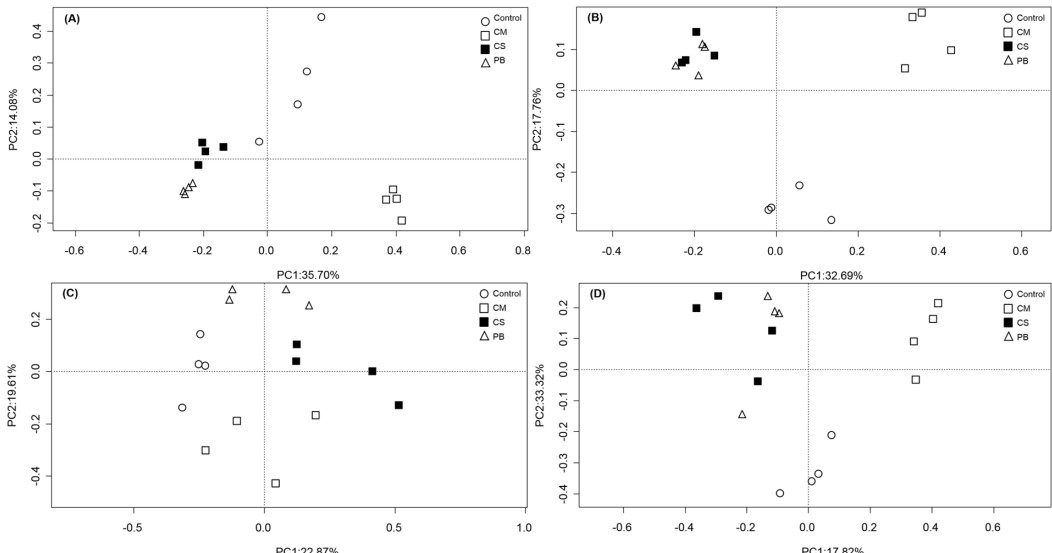

**Figure 1 Principal coordinate analysis (PCoA) of microbial community composition.** (A)bacterial community after 7 months, (B) bacterial community after 15 months, (C) fungal community after 7 months, and (D) fungal community after 15 months. CS, corn straw; CM, cow manure; PB, poplar branch.

community composition than the most common traditional organic amendment, CM. These differences were evident throughout the whole experimental period.

Distance-based RDA analysis showed that bacterial and fungal community compositions also were affected by different soil properties (Table 4). After seven months, pH, SOC, and TN were the factors that drove changes in both soil bacterial community composition (explaining 20.4, 9.7, and 14.0% of the total variance, respectively) and fungal community composition (explaining 13.1, 9.2, 11.4%, respectively). This result was similar to *Hartmann et al. (2015)*. However, AK also exhibited a strong effect on fungal community composition, accounting for 9.3% of the total variance. But among all soil parameters, pH was the

**Table 4  Correlations between soil microbial community composition and soil properties.** Values represent the estimation variance component (VC) that explained the distribution of microbial community composition are shown out of the bracket, and the corresponding levels of significance ($p$) are shown in the bracket. Values at $p < 0.05$ are shown in bold.

| Soil properties | 7 months | | 15 months | |
| --- | --- | --- | --- | --- |
| | Bacteria | Fungi | Bacteria | Fungi |
| pH | **20.45(0.001)** | **13.05(0.005)** | **12.83(0.008)** | **8.96(0.021)** |
| SOC | **9.73(0.024)** | **9.20(0.038)** | **10.12(0.012)** | **9.84(0.040)** |
| TN | **13.98(0.012)** | **11.44(0.012)** | **13.74(0.002)** | **9.53(0.041)** |
| TP | 5.49(0.195) | 5.31(0.420) | **14.30(0.003)** | **10.49(0.009)** |
| TK | 3.42(0.554) | 4.38(0.648) | 6.86(0.138) | 4.43(0.538) |
| AN | 6.18(0.152) | 6.62(0.209) | 5.34(0.221) | 5.07(0.374) |
| AP | 6.07(0.139) | 5.37(0.413) | 3.72(0.572) | 4.01(0.712) |
| AK | 6.43(0.130) | **9.34(0.038)** | 3.86(0.579) | **15.07(0.011)** |
| MBC | 3.93(0.419) | 4.30(0.681) | 3.85(0.550) | 3.58(0.737) |
| MBN | 3.92(0.496) | 4.52(0.670) | 3.19(0.733) | 4.21(0.625) |

strongest factor that drove variation in both bacterial and fungal community compositions in the soil; this was like the findings of other studies (*Lauber et al., 2009*; *Rutigliano et al., 2014*). Even after 15 months, pH, SOC, TN, and TP affected both bacterial (explained 12.8, 10.1, 13.7, and 14.3%, respectively) and fungal community composition (explained 9.0, 9.8, 9.5, 10.5, and 15.1%, respectively) in the soil. For fungal community composition, AK was the most influential factor, explaining 15.07% of the total variance. Therefore, we conclude that the fungal community in the desertified soils in our study were more sensitive to AK than the bacterial community. Moreover, we also noted that TP became a stronger factor affecting soil microbial community composition at the later stage. But, in the broadest perspective, pH, SOC and TN should be considered the most important factors driving microbial communities in this study, since they affected both bacterial and fungal communities throughout entire experimental period.

## Effects of local organic amendments on relative abundance of microbial phyla

Although local organic amendments were applied for a relatively short time in this study, we found, nevertheless, that distributions of microbial phyla in each treatment had changed significantly (ANOVA, $p = 0.001$ or <0.001) (Table S3). Moreover, we also found that all amendments induced changes in the abundance of some dominant bacterial and fungal phyla, compared to the Controls (Fig. 2A and Fig. 2B). In addition, changes in those dominant bacterial and fungal phyla among treatments were different between the two sampling times. This finding was also similar to *Zhang et al. (2018)*, and we assume that these changes through time were related to qualities of incorporated organic materials, changes in soil properties and sampling seasons. Also, similar to other studies (*Zeng, Dong & An, 2016*; *Lladó, López-Mondéjar & Baldrian, 2017*; *Zeng, An & Liu, 2017*), the dominant bacterial phyla of Proteobacteria, Actinobacteria, Acidobacteria, Bacteroidetes, and Firmicutes all were detected in our amended soil. Among all bacterial phyla, Proteobacteria was the most dominant and had higher abundance in treatments CS and PB, than in CM

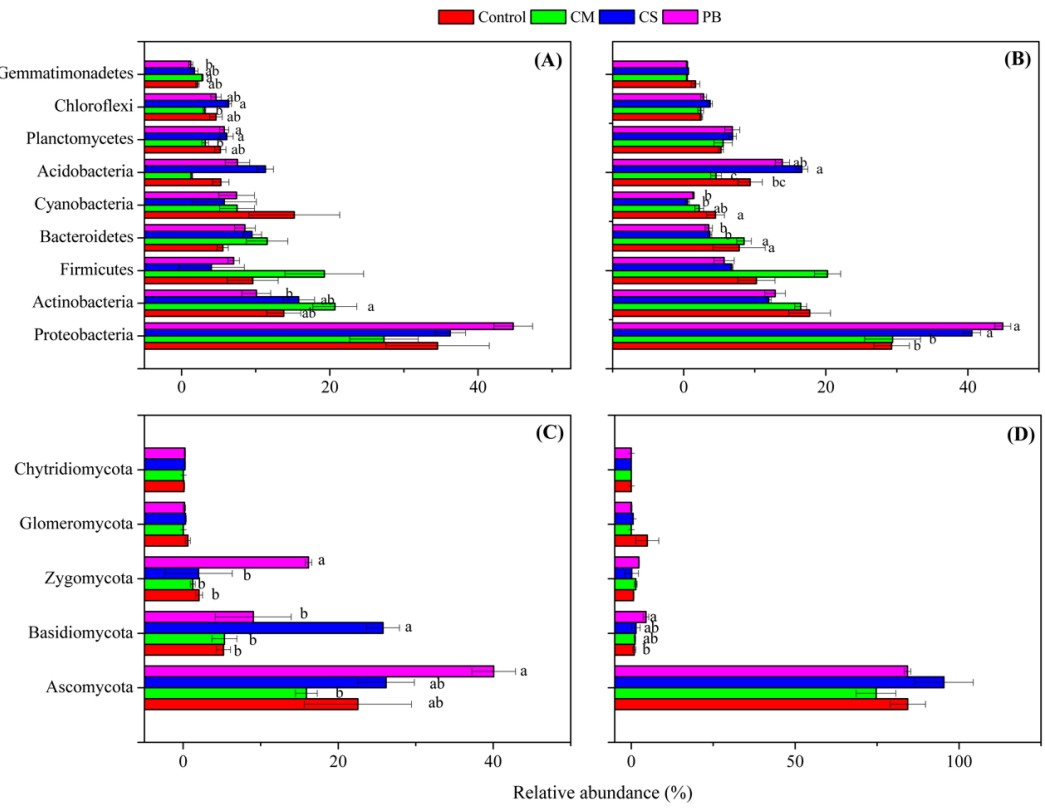

**Figure 2 Relative abundance of the dominant microbial phyla.** (A) bacterial phyla after 7 months, (B) bacterial phyla after 15 months, (C) fungal phyla after 7 months, and (D) fungal phyla after 15 months. Different letters indicate significant differences based on Tukey's HSD test ($p < 0.05$). CS, corn straw; CM, cow manure; PB, poplar branch. Only the phyla with significantly different relative abundance among amendments were labelled with letters. Error bars represent standard error ($n = 4$).

and the Control after 15 months ($p < 0.05$). This result might be due to the reason that Proteobacteria is copiotrophic microorganism (*Fierer et al., 2012*) and a decomposer of lignin (*Tian et al., 2015*). However, Proteobacteria was negatively correlated with TP and AN, but positively correlated with SOC and MBC after 15 months (Table 5), which was probably because CS and PB enhanced the microbial activity and fixed more nutrients resulting in microbial growth in the microcosm buckets after 15 months. This was supported by the much higher MBC we observed in CS and PB treatments than in CM (Table S2). Furthermore, previous researchers also documented that organic amendments with a high C:N ratio resulted in a higher microbial biomass and activity (*Heijboer et al.,2016*).

Similarly, Actinobacteria is also putatively identified as a group of copiotrophic taxawhich thrive in the condition of high C availability, and exhibit relatively rapid growth rates (*Fierer et al., 2012*). Our study supports this finding since the CM treatment, after 7 months, not only increased SOC and other soil nutrient contents (Table S2), but also increased abundance of Actinobacteria compared with the Control, (Fig. 2A). In addition, after seven months Actinobacteria was also positively correlated with AN, AP and AK, but negatively correlated with pH, (Table 5). Bacteroidetes is regarded as another group of

**Table 5  Correlations between relative abundance of major bacterial phyla and soil properties.** Bolded bacterial phyla were detected significantly different in relative abundance among treatments.

| Time | Bacterial phyla | pH | SOC | TN | TP | TK | AN | AP | AK | MBC | MBN |
|---|---|---|---|---|---|---|---|---|---|---|---|
| 7 months | Proteobacteria | 0.442 | −0.047 | −0.260 | −0.336 | −0.291 | −0.325 | **−0.515**[*] | −0.432 | 0.461 | 0.334 |
| | **Actinobacteria** | **−0.544**[*] | 0.198 | 0.409 | 0.349 | 0.009 | **0.448** | **0.589**[*] | **0.629**[**] | −0.449 | −0.187 |
| | Firmicutes | **−0.530**[*] | 0.202 | 0.400 | 0.432 | 0.221 | **0.540**[*] | **0.673**[**] | 0.411 | −0.453 | −0.407 |
| | Bacteroidetes | 0.393 | 0.326 | 0.363 | 0.360 | 0.457 | 0.154 | 0.304 | **0.604**[*] | 0.036 | 0.256 |
| | Cyanobacteria | −0.129 | −0.325 | −0.222 | −0.001 | 0.181 | 0.032 | −0.089 | −0.298 | −0.198 | −0.308 |
| | Acidobacteria | **0.554**[*] | −0.017 | −0.321 | **−0.616**[*] | −0.239 | **−0.800**[**] | **−0.718**[**] | −0.396 | **0.680**[**] | **0.571**[*] |
| | **Planctomycetes** | 0.352 | −0.192 | −0.452 | **−0.540**[*] | −0.352 | **−0.566**[*] | **−0.670**[**] | **-0.501**[*] | 0.416 | 0.348 |
| | **Chloroflexi** | 0.354 | 0.051 | −0.282 | −0.319 | −0.372 | **−0.566**[*] | **−0.576**[*] | −0.333 | 0.459 | 0.400 |
| | **Gemmatimonadetes** | −0.483 | −0.130 | 0.135 | 0.343 | 0.063 | **0.648**[**] | **0.615**[*] | **0.498**[*] | −0.495 | **−0.601**[*] |
| 15 months | **Proteobacteria** | −0.406 | **0.700**[**] | 0.342 | **−0.563**[*] | 0.218 | **−0.466** | −0.469 | −0.072 | **0.680**[**] | 0.000 |
| | Actinobacteria | 0.259 | **−0.587**[*] | −0.412 | 0.302 | 0.011 | 0.247 | 0.248 | −0.078 | **−0.566**[*] | −0.031 |
| | Firmicutes | −0.158 | −0.123 | 0.232 | **0.823**[**] | 0.217 | **0.817**[**] | **0.819**[**] | 0.491 | −0.400 | 0.255 |
| | **Bacteroidetes** | 0.238 | −0.235 | −0.060 | 0.440 | −0.404 | 0.356 | 0.393 | 0.181 | −0.304 | −0.135 |
| | **Cyanobacteria** | **0.578**[*] | −0.400 | −0.467 | 0.172 | **−0.569**[*] | −0.048 | 0.022 | −0.216 | **−0.585**[*] | 0.015 |
| | **Acidobacteria** | −0.031 | 0.164 | −0.119 | **−0.814**[**] | 0.104 | **−0.718**[**] | **−0.753**[**] | −0.419 | 0.478 | −0.003 |
| | Planctomycetes | 0.089 | 0.052 | 0.163 | −0.149 | 0.137 | −0.104 | −0.164 | 0.048 | 0.303 | 0.111 |
| | Chloroflexi | −0.329 | 0.178 | 0.179 | **−0.502**[*] | 0.299 | −0.243 | −0.339 | −0.218 | 0.489 | −0.199 |
| | Gemmatimonadetes | **0.652**[**] | **−0.644**[**] | **−0.646**[**] | −0.125 | **−0.665**[**] | −0.307 | −0.270 | −0.393 | −0.480 | −0.438 |

**Notes.**
[*] $p < 0.05$.
[**] $p < 0.01$.
typically copiotrophic taxa (*Fierer, Bradford & Jackson, 2007*), and have been found to be positively correlated with soil total P (*Tian et al., 2015*) and soluble P (*Yashiro et al., 2016*). In our study, Bacteroidetes had higher abundance in CM than in CS and PB treatments after 15 months (Fig. 2B), but we did not find any soil nutrient that was correlated with this group (Table 5). Hence, we speculate that Bacteroidetes is slow-growing copiotrophic microorganism (*Fierer, Bradford & Jackson, 2007*), which reacted to soil nutrients too slowly to detect in our short-term study.

Though after 15 months, Cyanobacteria also showed comparatively higher abundance in the treatment CM than CS and PB (but not significantly different from each other) (Fig. 2B), this group is regarded as a $N_2$-fixing microorganism common in soils and soil crusts (*Antoninka et al., 2016*; *Rippin et al., 2018*). Therefore, we assume that the higher values of TN and AN in CM relative to CS and PB could be the reason for higher abundance of Cyanobacteria found in CM (Table S2), although our correlation analysis did not detect this (Table 5). Nevertheless, Acidobacteria is regarded to be more adapted to a nutrient-limited soil environment (*Ward et al., 2009*; *Fierer et al., 2012*) and some scientists also characterize them as generally preferring soil environments with high carbon resource (*Voříšková & Baldrian, 2013*). Our study also reinforces that viewpoint because after 15 months Acidobacteria was found in higher abundance in treatments CS and PB (Fig. 2B), which contained lower available N, P, and K at that sampling time (Table S2). Furthermore, Acidobacteria was also negatively correlated with TP, AN, and AP in this study (Table 5). Gemmatimonadetes also showed copiotrophic features in this study, because a greater relative abundance was detected in the higher-nutrient treatment CM after seven months (Fig. 2A). Correlation analysis also revealed that the relative abundance of Gemmatimonadetes was positively correlated with AN, AP, and AK (Table 5). Similarly, some previous studies support our findings (*Bernard et al., 2007*; *Pfeiffer et al., 2013*), yet little is known about the ecological roles of Gemmatimonadetes (*Whitman et al., 2016*). In contrast, both Planctomycetes and Chloroflexi were more abundant in treatments CS and PB than in CM after 7 months (Fig. 2A). Similarly, correlation analysis also indicated that those two bacterial phyla were negatively correlated with some soil nutrients (Table 5). However, we assume that Planctomycetes were related to decomposition of CS and PB in soils, since Planctomycetes are regarded as the decomposers of organic matter, including cellulose and hemicellulose (*Chapman et al., 2017*). The high abundance of Chloroflexi may be a result of the accumulation of halogenated organic compounds generated during decomposition (*Chapman et al., 2017*).

Fungal phyla compositions were also different among treatments between the two sampling times (Fig. 2C and Fig. 2D). Though we know little about the ecological function of detected fungal phyla, some studies reported that the cellobiohydrolase gene is unique to fungi, broadly distributed in Ascomycota and Basidiomycota in forest soils, and encodes for an enzyme which is critical to cellulose breakdown (*Edwards et al., 2011*). Therefore, we assume that the higher relative abundance of Ascomycota (after 7 months), Basidiomycota (after both 7 and 15 months), and Zygomycota (after 7 months) detected in treatments PB and CS (Fig. 2C and Fig. 2D) could be correlated with the process of wood and straw decomposition in our treatments. Correlation analysis also showed that Ascomycota was

Li et al. (2019), *PeerJ*, DOI 10.7717/peerj.6854

**Table 6 Correlations between relative abundance of fungal phyla and soil properties.** Bolded fungal phyla were detected significantly different in relative abundance among treatments.

| Time | Fungal phyla | pH | SOC | TN | TP | TK | AN | AP | AK | MBC | MBN |
|---|---|---|---|---|---|---|---|---|---|---|---|
| 7 months | **Ascomycota** | 0.425 | 0.279 | −0.205 | −0.284 | 0.025 | **−0.515**[*] | −0.483 | −0.454 | **0.665**[**] | 0.382 |
| | **Basidiomycota** | 0.479 | 0.323 | 0.020 | −0.090 | −0.013 | **−0.588**[*] | −0.384 | 0.055 | **0.674**[**] | **0.610**[*] |
| | **Zygomycota** | 0.322 | 0.214 | −0.090 | −0.225 | −0.052 | −0.421 | −0.342 | −0.383 | 0.408 | 0.458 |
| | Glomeromycota | 0.257 | **−0.527**[*] | −0.411 | −0.214 | −0.146 | −0.320 | −0.421 | −0.427 | −0.107 | −0.172 |
| | Chytridiomycota | 0.284 | −0.028 | −0.260 | −0.284 | 0.025 | **−0.636**[**] | −0.483 | −0.454 | 0.394 | 0.494 |
| 15 months | Ascomycota | 0.096 | 0.183 | −0.137 | −0.408 | −0.023 | −0.463 | −0.492 | **−0.598**[*] | 0.234 | 0.200 |
| | **Basidiomycota** | −0.155 | 0.423 | 0.153 | −0.231 | −0.172 | −0.223 | −0.241 | 0.062 | 0.331 | −0.167 |
| | Zygomycota | −0.140 | 0.065 | 0.133 | −0.013 | −0.081 | 0.070 | 0.083 | **0.541**[*] | 0.070 | −0.217 |
| | Glomeromycota | 0.356 | **−0.725**[**] | **−0.634**[**] | −0.129 | −0.179 | −0.238 | −0.216 | −0.412 | **−0.509**[*] | −0.321 |
| | Chytridiomycota | 0.175 | −0.188 | −0.243 | −0.332 | −0.441 | −0.300 | −0.280 | 0.142 | −0.026 | −0.482 |

Notes.
[*] $p < 0.05$.
[**] $p < 0.01$.
positively correlated with MBC, and Basidiomycota was positively correlated with MBC and MBN, but both were negatively correlated with AN (Table 6). Hence, those fungal phyla might be accelerated to fix more AN and to decompose incorporated PB and CS in soils.

In addition, we consider the above fungi to play an essential role in incorporated organic material decomposition in the early stages, since some studies indicated that fungi were dominant litter decomposers at an early stage (*Voříšková & Baldrian, 2013*; *Žifčáková et al., 2016*), while bacteria were dominant decomposers at later stages (*Berg, 2014*). However, it is essential to acknowledge that we can only speculate on the ecological role of the detected taxa based on what has been previously described in other studies. Also, we only identified several organic amendment-sensitive bacteria and fungi to the level of phylum in this study. Therefore, a more in-depth analysis should also be conducted based on our data to find more information about specific microbial ecological roles. Moreover, long-term field studies should also be considered as the next important step. Importantly, we should acknowledge that alfalfa growth also likely affected the microbial community, although our main objective was to tease out the effects of different organic materials as soil amendment strategies.

## CONCLUSIONS

In this study, we found that soil microbial $\alpha$-diversity reacted weakly to all local organic amendments. However, bacterial and fungal community compositions changed significantly for different amendments. Non-traditional organic amendments such as woodchips (treatment PB) exhibited similar effects to corn straw treatment (CS). Both PB and CS treatments influenced soil communities differently that the most traditional amendment using cow manure (CM). Soil organic carbon, total nitrogen and pH were the most important factors driving microbial community structure. In conclusion, woodchips should also be considered as a viable soil amendment for future applications.

## ACKNOWLEDGEMENTS

We thank Drs. Xilu Ni, Changxiao Li and Jian Li for helping experimental design and data collection, and Dr. Tsafack Noelline for helping data analysis.

### Funding

This work was financially supported by the China Postdoctoral Science Foundation (2018M643771) and the Top Discipline Construction Project of Pratacultural Science of Ningxia University (NXYLXK2017A01). The funders had no role in study design, data collection and analysis, decision to publish, or preparation of the manuscript.

### Grant Disclosures

The following grant information was disclosed by the authors:

China Postdoctoral Science Foundation: 2018M643771.
Top Discipline Construction Project of Pratacultural Science of Ningxia University: NXYLXK2017A01.

## Competing Interests

The authors declare there are no competing interests.

## Author Contributions

- Zhigang Li and Kaiyang Qiu conceived and designed the experiments, performed the experiments, analyzed the data, prepared figures and/or tables, authored or reviewed drafts of the paper, approved the final draft.
- Rebecca L. Schneider and Stephen J. Morreale conceived and designed the experiments, analyzed the data, approved the final draft.
- Yingzhong Xie conceived and designed the experiments, contributed reagents/materials/analysis tools, approved the final draft.

## Data Availability

The raw measurements are available in the Supplemental Files.

## Supplemental Information

Supplemental information for this article can be found online at http://dx.doi.org/10.7717/peerj.6854#supplemental-information.

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
