# Peer review of "Comparison of microbial community structures in soils with woody organic amendments and soils with traditional local organic amendments in Ningxia of Northern China"

_PeerJ, doi:10.7717/peerj.6854_

## Round 0.1 · original submission · Major Revisions

Please, improve the English thorough the ms accordingly to reviewer 1 suggestions. Consider to include more information about the state of the art related to soil microbial communities. Also, consider to present the results together with the discussion in one section, taking care of improving the discussion of the main results with the available bibliography. Re-write the conclusions and review all the references both in the text and in the list of references.

·

Basic reporting

The English used in the ms is acceptable for its understanding although some corrections are recommended
Introduction should include more information on the state-of-the-art of microbial communities and soil properties. This information can also be used to help in the discussion of the ms.
The ms is difficult to read. Perhaps, most of the Results section could be removed because it is summarized in the Discussion. The Discussion, after selecting the results in it, is quite limited and should be improved. Similarly, the provided Conclusions are mostly a repeated text of Results and should be more elaborated.
This is a descriptive ms with provide data on microbial communities and soil properties at two time-periods (7 and 15 months) for several soil treatments using different sources for the amendments.

Experimental design

The topic of the ms fits within the Aims and Scope of the Journal.
The objective of the ms is the comparison of different soil amendments on microbial communities and soil properties. Nevertheless, it is not well defined how the provided results fills current knowledge gaps. The ms present some data which are potentially of interest but they are poorly analyzed and no meaningful conclusions are clear.
Methods are described but some improvements can be appropriate. Most characteristics of soil and amendments could be provided in a Table so that the text can be reduced and simplified. It is not clear the quantity of soil and the amendments mixed in the treatments; for instance, does the weight of soil corresponds to dry weight?.
I have a major methodological concern to this ms. The addition of a N and P source to soil and amendments. I understand that supplementing soil with different amendments is aimed to maintain or improve soil nutrients and properties so that plant growth is enhanced and soil impoverishment is avoided. However, the authors add a N (urea) and P (KH2PO4) source to their treatments. The addition of extra N and P sources besides the amendments could hide the effect of different amendments on soil properties and above all on microbial communities because potential restrictions on these nutrients might be masked.
The authors mention "pyrosequencing" in lines 103 and 205. However, they do not use this technique, rather they use Illumina sequencing (line 226). Please, correct because the Illumina platform does not perform pyrosequencing.
Please, avoid reporting Bray-Curtis distance and mention Bray-Curtis dissimilarity which is the correct way. This is an approach to beta-diversity which is not mentioned in the ms.
Please, avoid reporting Bray-Curtis distance and mention Bray-Curtis dissimilarity which is the correct way. This is an approach to beta-diversity which is not mentioned in the ms.
Please, correct OTU throughout the ms instead of OUT.
Statistical procedures do not look to appropriated due to the inconsistencies reported by the authors repeatedly throughout the ms.

Validity of the findings

The final outcome of the performed comparison is far from being clear. The data are there but the authors should analyze them more in depth. Perhaps, different, more appropriated statistical treatments are needed, perhaps some methodological factors contribute to mask (see above) clear effects of the amendments. It would be desirable to provide some meaningful conclusions to the author’s set of data beyond a simple (difficult to read) description.
In order to improve the readability of the ms, I would recommend to simplify the text and report in writing the important observations which will contribute to important discussions and conclusions. For example, most of the Results (correct Results and not Rusults) could be deleted. Most of the important stuff is included (repeated) in Discussion. Removing the Results, and marking Results and Discussion the current Discussion, the ms will be much more readable. The Discussion is also awaiting for improvements. Based on a more in depth Introduction, the Discussion could built up and add new knowledge to the state-of-the-art providing information beyond a simple description of results. A more informative Conclusions section could be written. The statistical analysis of the results does not look to appropriated judging for the “inconsistencies” that the author report repeatedly as pointed by the authors (line 365). Perhaps, the authors, after a detailed analysis, could provide with explanation for those data. An advice would be to simplify the results; for instance, the use of one diversity index would make the analysis easier than using four different indices.
Care must be taken when approaching potential physiological features to 16S rRNA sequencing data. Cyanobacteria should be considered, at first, as primary producer (photosynthetic) microorganism common in soils rather than as a copiotroph (line 447). A priori, one should expect that the availability of inorganic N and P sources might rule their development.
A major result of this set of data could be that pH, SOC and TN could be the most important parameters driving microbial communities during the amendments of soil (lines 399-400).
Tables 4, 5 and 6. What is according to the authors the difference among Relationships and Correlations?
Figure 1 shows differential distribution of treatments but Figure 2 shows no major differences among the different phyla for each treatment. The authors should consider if these differences are significant.

Additional comments

The authors should try to analyze in detail their results in accordance with current state-of-the-art so that informative conclusions, adding up to current knowledge, can be provided. Different potential suggestions have been provided above.

·

Basic reporting

This manuscript is an original research paper investigating the impact of different soil amendments on the diversity and community structure of soil microbes (Bacteria and fungi), and the soil chemical properties that shape microbial diversity and community structure. Fundamentally the study is solid and well design. The authors should revise the reference list carefully, cross-check with the text as some reference is cited in the text and missing in the reference list. The referencing style should be checked carefully in-text as well as in the reference list and corrected according to the journal instruction to authors. The spelling for some author’s names in the reference list is not correct. In general, the manuscript is well written. However, I have raised some minor issues that the authors should address.

Experimental design

Experimental design is solid and well done. The multivariate statistic methods and statistical tests used are suitable for the study. I’ve got no criticism in this regard.

Validity of the findings

Besides evaluating the influence of amendments on the diversity and community structure of the soil microbial diversity at two point times (after seven weeks, and after fifteen weeks), the authors compare the influence of two local organic amendments (cow manure, corn straw) versus a woody organic amendment (poplar branches) on soil microbial community structure. Very little works have been done previously on that aspect in the field of microbial ecology. The study is, therefore, important as it is a welcome contribution that advances the field of study, and contributes to the body of knowledge. In general, the manuscript is well written. However, I have raised some minor issues that the authors should address.

Additional comments

The authors should revise the reference list carefully, cross-check with the text as some reference is cited in the text and missing in the reference list. The referencing style should be checked carefully in-text as well as in the reference list and corrected according to the journal instruction to authors. The spelling for some author’s names in the reference list is not correct. In general, the manuscript is well written. However, I have raised some minor issues that the authors should address.

Introduction

Line 75: Please, check the year of publication for the citation. It is listed as (Fernández-Gálvez et al., 2009) in the text, but as “2012” in the reference list (line: 529).

Line 85: Please, check the year of publication for the citation. It is listed as (Dai and Chu, 2012) in the text, but as “2010” in the reference list (line: 519). Also, replace “and” with “&” in the citation of interest.

Line 87: The citation (Jia et al., 2013) must be in italic

Lines 99-100: The citation (Barrios, 2007) is not listed in the reference list

Material & methods

Line 147: add authority to “Populus alba”.
Line 148: add authority to “Zea mays”.

Line 196: Please, check the year of publication for the citation. It is listed as (Nelson et al., 1982) in the text, but as “1996” in the reference list (line: 577).

Line 201: Replace “and” with “&” in the citation “(Tran and Simard, 1993)”

Line 213: The citation (Weisburg et al., 1991) must be in italic

Line 215: The reference must be cited as “(Gardes & Bruns, 1993)”

Line 239: “(Edgar, 2013)” must be in italic

Line 242: OTU and not “OUT.”

Line 283: Please, clarify the meaning of “CM was also separated from CM”

Lines 284 and 284: Please, indicate clearly if it is either (Fig. 1a) or (Fig. 1b), or both?

Line 316: I suggest that the output result of db-RDA be represented by a “biplots” as well, showing the soil chemical properties that significantly influence the bacterial and fungal communities, respectively. The plots of interest can be made in “vegan” after the analysis.

Discussion

Lines 357 and 358: Add a “comma” after “ et al.” to the citations (Francioli et al.,2016; Sharma et al., 2017)

Line 366: Replace “and” with “&” in the citation “(Hartmann and Widmer, 2006)”

Line 367: OTU and not “OUT”

Line 368: The citation “Calleja-Cervantes et al. (2015)” is not listed in the reference list

Lines 404 and 405: Add a “comma” after “ et al.” to the citations (Lauber et al., 2009; Rutigliano et al., 2014)

Line 423: Add a “comma” after “ et al.” to the citations (Zeng et al., 2016; Lladó et al., 2017; Zeng et al., 2017)

Line 428: The year of publication for the citation “(Tian et al., 2014)” is “2014” in the text, but “2015” in the reference list (line: 599)

Line 441: The same citation is written as “(Tian et al. 2015)”. Please, check!

Line 442: Add a “comma” after “ et al.” in “(Yashiro et al., 2016)”

Lines 452 and 471: Replace “and” with “&” in the citation “(Větrovský and Baldrian, 2013)”

Line 472: The citation (Berg, 2014) is not listed in the reference list

Line 513: REFERENCES and not “EFERENCES.”

Reference List

Line 514: “Caporaso et al.” was not cited in the text and should be deleted in the reference list

Line 519: Replace "and" with a "comma" in “Dai G and Chu W. 2010”

Line 520: The journal’s name “Forest Resources Management” must be in italic

Line 521: Add a full stop after “2013” in “Edgar RC. 2013”

Line 529: “2009” or “2012”? Please, check the year of publication carefully

Line 533: Add a full stop after "RB" in “Fierer N, Bradford MA, Jackson RB 2007.”

Line 577: “1982” or “1996”? Please, check the year of publication carefully

Line 608: Remove the brackets around “2006.”

Line 613: The Author's name “Voříšková” is misspelt either in the text (line 452) or in the reference list. Please check!

---

## Round 0.2 · Minor Revisions

Please include the suggested changes od reviewer 1 and resubmit the manuscript for its publication.

·

Basic reporting

The ms has improved considerably and it is almost ready for publication

Experimental design

Experimental design has been reexplained and it is satisfactory

Validity of the findings

A great improvement is observed in the last version of the ms

Additional comments

Just two points for the authors to improve in their ms:
- lines 258-259: Please, correct: ... 10,510 sequences were considered per bacterial community and ...
- Conclusion. lines 437-458. The conclusions sections can be significantly reduced. for example:
Bacterial and fungal community compositions changed significantly for different amendments. Non-traditional organic amendments such as woodchips (treatment PB) exhibited similar effects to corn straw treatments (CS). Both PB and CS treatments influenced soil communities differently that the most traditional amendment using cow manure (CM). Soil organic carbon, total nitrogen and pH were the most important factors driving microbial communty structure. Woodchips should also be considered as a viable soil amendment for future applications.

---

## Round 0.3 · accepted · Accept

I believe that your manuscript now meets the requirements to be published in PeerJ.